# Recent Advances in Characterization of Melanin Pigments in Biological Samples

**DOI:** 10.3390/ijms24098305

**Published:** 2023-05-05

**Authors:** Kazumasa Wakamatsu, Shosuke Ito

**Affiliations:** Institute for Melanin Chemistry, Fujita Health University, Toyoake 470-192, Aichi, Japan

**Keywords:** eumelanin, pheomelanin, alkaline hydrogen peroxide oxidation (AHPO), hydroiodic acid hydrolysis, Raman spectroscopy, electron paramagnetic resonance spectroscopy, pump–probe microscopy, LC–MS, tape stripping

## Abstract

The melanin pigments eumelanin (EM) and pheomelanin (PM), which are dark brown to black and yellow to reddish-brown, respectively, are widely found among vertebrates. They are produced in melanocytes in the epidermis, hair follicles, the choroid, the iris, the inner ear, and other tissues. The diversity of colors in animals is mainly caused by the quantity and quality of their melanin, such as by the ratios of EM versus PM. We have developed micro-analytical methods to simultaneously measure EM and PM and used these to study the biochemical and genetic fundamentals of pigmentation. The photoreactivity of melanin has become a major focus of research because of the postulated relevance of EM and PM for the risk of UVA-induced melanoma. Our biochemical methods have found application in many clinical studies on genetic conditions associated with alterations in pigmentation. Recently, besides chemical degradative methods, other methods have been developed for the characterization of melanin, and these are also discussed here.

## 1. Introduction

The term “melanin” was first coined by Berzelius in 1840 to refer to black animal pigments [1]. Since then, it has been widely used to refer to any black or dark brown organic pigment. Nicolaus suggested the classification of melanins into three groups: eumelanin (EM), pheomelanin (PM), and allomelanins—the latter being a class of dark non-nitrogenous pigments of plant, fungal, or bacterial origin [2]. d’Ischia et al. (2013) proposed to use the term “melanin” only for pigments of diverse structure and origin derived from the oxidation and polymerization of tyrosine in animals or phenolic compounds in lower organisms [1]. EM is a black-to-brown, insoluble subgroup of melanin pigments derived at least in part from the oxidative polymerization of L-3,4-dihydroxyphenylalanine (DOPA) via 5,6-dihydroxyindole (DHI) and 5,6-dihyroxyindole-2-carboxylic acid (DHICA) intermediates [3,4]. PM is a yellow-to-reddish-brown, alkali-soluble, sulfur-containing subgroup of melanin pigments derived from oxidation of cysteinyldopa (CD) precursors via benzothiazine (BT) and benzothiazole (BZ) intermediates [5]. Neuromelanins (NMs) are dark pigments produced within neurons by oxidation of dopamine (DA), norepinephrine (NE), or other catecholaminergic precursors [6,7,8]. Pyomelanins are dark pigments produced mainly, but not exclusively, by microorganisms from homogentisate [9]. Melanin pigments produced by oxidation of DA are universally found on the body surface of insects [10].

Melanin pigments in vertebrates can be classified into two basic groups of pigments: EM and PM. Both EM and PM are highly oxidized and complex polymers. The biochemical pathway of melanin formation has been described in detail in our previous review articles [1,3,4,11]. A summary of the biochemical pathways to EM and PM is shown in Figure 1. Both EM and PM are derived from the common precursor dopaquinone (the *ortho*-quinone of L-DOPA, DQ) that is produced from L-tyrosine by the action of the melanogenic enzyme tyrosinase. Conversion of tyrosine to DQ by tyrosinase is the first critical step in melanogenesis [12,13,14]. DQ is a highly reactive *ortho*-quinone intermediate that reacts exceptionally rapidly with thiol compounds such as cysteine or glutathione [14,15]. In the absence of thiol compounds, DQ undergoes intramolecular cyclization of its amino group to produce dopachrome (DC) via cyclodopa (or leucodopachrome) [16]. Cyclodopa is then rapidly oxidized by a redox reaction with DQ to give DC and L-DOPA [16] (Figure 1). DC is then spontaneously and gradually converted mostly to DHI by decarboxylative rearrangement and to a lesser extent to DHICA [17,18,19], the ratio of which is determined by another melanogenic enzyme called DC tautomerase (Dct, tyrosinase-related protein-2, Tyrp2) [20,21,22]. Dct and copper ions can catalyze the tautomerization of DC to DHICA [20,21,22,23]. Before the *Tyrp2* locus was characterized, Dct was discovered as a novel enzyme in melanogenesis [20,24,25,26,27,28]. EM is produced from various ratios of DHI and DHICA by oxidative polymerization. Oxidation of DHI is catalyzed directly by tyrosinase or indirectly by DQ, while oxidation of DHICA appears to be catalyzed by Tyrp1, the brown locus protein, at least in mice [29,30]. However, the precise enzymatic function of the human homolog TYRP1 is not yet clear, and it may not act as it does in mice [31]. In the presence of L-cysteine, DQ preferentially yields cysteinyldopa (CD) isomers. Production of PM appears to proceed spontaneously after the production of DQ as long as cysteine is present to produce CDs [12,32]. Oxidation of CDs proceeds by redox exchange with DQ to produce the quinone form. Cyclization and rearrangement create BT intermediates that are oxidized to form BZ intermediates and then BT and BZ intermediates are gradually co-polymerized to form PM with both BT and BZ moieties [5] (Figure 1).

The quantity and ratio of EM and PM are the major determinants for the color of hair, skin, and eyes [34,35,36,37]. Furthermore, it is generally accepted that EM is photoprotective, while PM is phototoxic to tissues [38]. Therefore, in studies about melanin, it is pivotal to measure the quantity and the ratio of EM and PM. Fortunately, enabling such measurement by deductive analysis, several specific degradation products from EM and PM were identified by extensive chemical studies by Nicolaus, Prota, and their associates in Naples in the 1960s [2,39,40]. Among others, it is noteworthy that acidic KMnO_4_ oxidation of EM yields, as a major pyrrolic degradation product, pyrrole-2,3,5-tricarboxylic acid (PTCA) [35,36,40,41], while hydroiodic acid (HI) reductive hydrolysis of PM yields an isomeric mixture of 4-amino-3-hydroxyphenylalanine (4-AHP) and 3-amino-4-hydroxyphenylalanine (3-AHP), which derive form the oxidative polymerization of 5-*S*-cysteinyldopa (5SCD) or 2-*S*-cysteinyldopa (2SCD), respectively [34,42] (Figure 1). Taking advantage of this information, in 1983, Ito and Jimbow introduced an HPLC-based method to quantify EM and PM in tissue samples, which made the isolation of melanin unnecessary [42]. However, the method required relatively large amounts of samples and was time-consuming. Therefore, in 1985, the method was improved by Ito and Fujita with respect to rapidness, simplicity, and sensitivity [34]. The acidic KMnO_4_ oxidation of EM was used widely until 2011 [43] when we developed a more convenient method using alkaline hydrogen peroxide oxidation (AHPO) to quantify EM and PM simultaneously.

Compared to previous reviews on methods of melanin analysis [44], this review (i) more comprehensively summarizes the usefulness of AHPO and related methods for analyzing EM and PM in various tissue samples including fossil specimens, (ii) describes the application of our methods to study photodegradation of EM and PM, and (iii) introduces recent advances in analyzing melanins using various physicochemical methods, some of which are non-invasive and non-destructive.

## 2. Microanalytical Applications of the Chemical Degradation of Melanin

Most natural melanin pigments are composed of both EM and PM (the concept of “mixed melanogenesis”) [4,45]. We developed a microanalytical method to analyze EM and PM based on chemical degradation of melanin pigments followed by the analysis of degradation products using high-performance liquid chromatography (HPLC) with UV detection for AHPO or electrochemical detection for HI hydrolysis [3,34,35,36] (Table 1). Recently, we developed a more convenient method to measure EM and PM simultaneously [43,46,47,48] (Figure 2).

### 2.1. Analysis of EM and PM Using Chemical Degradation Followed by HPLC

The AHPO method yields the specific markers PTCA, pyrrole-2,3-dicarboxylic acid (PDCA), thiazole-2,4,5-tricarboxylic acid (TTCA), and thiazole-4,5-dicarboxylic acid (TDCA). PTCA and PDCA are specific biomarkers for DHICA units or 2-substituted DHI units, and for DHI units, respectively. TTCA and TDCA are biomarkers specific for the BZ-derived units of PM. In addition to PTCA and PDCA, pyrrole-2,3,4,5-tetracarboxylic acid (PTeCA) and pyrrole-2,3,4-tricarboxylic acid (isoPTCA) were also detected in fossil ink sacks [1,49] suggesting that the DHI units are additionally cross-linked during the aging of EM polymers. The PTeCA/PTCA ratio has been proposed as a good indicator of EM aging [50].

AHPO was developed to overcome the disadvantages of the acidic KMnO_4_ oxidation method (Table 1) [43,44,45,46]. Namely, the procedure of acidic KMnO_4_ oxidation is rather complex and involves ether extraction, and its yield of PDCA is too low for usage as a specific marker for DHI-derived EM such as DA melanin (Table 1). These shortcomings were mostly overcome by AHPO, because this technique omits the tedious ether extraction, allowing a direct injection to HPLC and an analysis of PDCA in reasonable yields in addition to PTCA. We compared the PTCA levels obtained with AHPO to those with KMnO_4_ oxidations in 38 human hair samples and found an excellent correlation (r = 0.947) with a slope of 2.00 (Figure 3). This indicates that AHPO is as reliable as KMnO_4_ oxidation for analyzing EM using its specific degradation marker PTCA.

The following are examples of studies that used PDCA or TTCA determination:

We applied AHPO to analyze the DHI/DHICA ratio caused by *Dct* mutation [51]. The slaty (*slt/Dct^slt^*) mutation is known to reduce the activity of DC tautomerase that converts DC to DHICA in eumelanogenesis [22]. Hirobe et al. (2015) showed that DHICA content (23%) estimated by the PDCA/PTCA ratio in slaty hairs was significantly lower than in black hairs (>50%) [51].

We applied AHPO to investigate BZ moieties in PM. EM was generally believed to account for more than 90% of TM, but studies by del Bino et al. found that human epidermis contains 74% EM and 26% PM, with PM being mostly of the BZ type [52,53].

Ito et al. (2018) developed a new method for melanin characterization based on HCl hydrolysis and subsequent AHPO of tissue samples [47]. This method removes proteins and low-molecular-weight compounds from tissue samples, resulting in a more simplified HPLC chromatogram compared to conventional AHPO. This study concluded that human dark (brown to black) hair melanin consists of approximately 85% EM and 15% BZ-derived PM, as indicated by the calibration curve for TDCA/PDCA ratios after HCl hydrolysis against melanins prepared from various ratios of DOPA and cysteine [47]. Ito et al. (2011) initially hypothesized that TTCA and TDCA in AHPO product of dark human hair artificially derive from protein-bound DOPA and CD [43], but the new method based on HCl hydrolysis demonstrated that TTCA and TDCA in AHPO product of dark human hair come, in fact, from BZ-derived PM. Thus, it was found that a low but constant level of BZ-derived PM is present in eumelanic black to brown human hair. It was also found that upon acid hydrolysis of EM, the DHICA moieties undergo decarboxylation to form cross-linked EM via DHI moieties. BZ–PM moieties also undergo decarboxylation with conversion of TTCA to TDCA. Another advantage of the HCl–AHPO is that the HCl hydrolysis removes mineral components in fossil specimens (demineralization), which led to the detection of EM markers that were hardly detectable using AHPO alone [54].

Initially, our HPLC conditions (0.1 M phosphate buffer:methanol = 99:1) for analyzing PTCA and other markers encountered some problems such as a short life of the HPLC column and interference from a preceding sample [48]. Additionally, with the exception of PTCA as a major EM marker, trace levels of other markers sometimes overlapped with interfering peaks. These problems could be overcome by adding anionic ion-pair reagents such as tetra-*n*-butylammonium bromide (1 mM). The methanol concentration was increased from 1% in the original method to 17% (from 15% to 30% for PTeCA), and the improved method showed good marker linearity, reproducibility, and recovery. Figure 4 shows the improved HPLC chromatograms of the melanin markers and the AHPO mixtures from human black hair, human red hair, and dark human epidermis. PTCA and TTCA, the major products from EM and PM, were well separated from other interfering peaks. Using this improved HPLC method, we compared the original method with the improved one using synthetic melanin, mouse hair, human hair, and human epidermal samples. In addition to PTCA, also TTCA—a major marker of the BZ moiety of PM—showed an excellent correlation between both HPLC methods [48]. Peaks for other markers such as PDCA became better distinguishable because of the attenuation of interfering peaks. This improved HPLC method is recommended for quantitative analysis of melanin markers in AHPO because of its simplicity, accuracy, and reproducibility (Table 2).

The improved AHPO [48] has made it possible to analyze PDCA more accurately as well. Del Bino et al. (2022) could quantify DHI melanin in human skin by comparing the PDCA/PTCA ratios (and the PTCA/A500 ratios; see below) against the calculation curves for melanins prepared from various ratios of DHI and DHICA [53]. A good correlation was obtained between the total melanin (TM) amount obtained using spectrophotometry and the TM amount obtained using HPLC after chemical degradation. The calculated EM and PM proportions in human epidermis melanin were 74% and 26%, respectively. Furthermore, the calculated proportions of DHI, DHICA, BZ, and BT in human skin melanin were 35%, 41%, 20%, and 4%, respectively. The ratios of EM to PM and DHI melanin to DHICA melanin in skin were constant regardless of the degree of pigmentation [53].

As a similar approach to improve the HPLC conditions, Panzella et al. (2006) reported a simultaneous chromatographic methodology after AHPO by the use of an octadecylsilane column with polar end-capping on HPLC with 1% formic acid (pH 2.8)/methanol as the eluent to separate PTCA for EM, and TTCA and 6-(2-amino-2-carboxyethyl)-2-carboxy-4-hydroxybenzothiazole for PM [55].

After we first introduced the analytical method using HI hydrolysis, we used the combined amount of two isomers of AHPs (4-AHP and 3-AHP) as a marker of pheomelanin [34]. However, one problem with using the total AHPs as a marker was that background levels originated from precursors other than pheomelanin. In order to better solve this problem, we developed HPLC conditions which allow good separation of 4-AHP and 3-AHP, which are degradation products of BT-derived moieties arising from 5SCD and 2SCD in PM, respectively [46]. It was found that considerable and variable amounts of background 3-AHP are produced from other sources than PM, most likely 3-nitrotyrosine residues in proteins.

Recently, attempts to analyze melanin markers using a more specific method have been reported in combination with AHPO. Petzel-Witt et al. (2018) used liquid chromatography–mass spectrometry (LC–MS/MS) to determine PTCA in hair samples [56]. Three transitions in PTCA fragmentation (*m*/*z* 198 [M–H]^−^, 154, 110, 66) were monitored, and the method was also applied in another study [57,58]. Lerche et al. (2019) reported a novel LC–MS/MS method to quantify EM and PM and their relation to UVR sensitivity [59]. Additionally, solid-phase extraction (SPE) has been introduced to purify the AHPO mixtures prior to HPLC. Rioux et al. (2019) reported a method to measure EM and PM in melanoma tumors and cells using SPE after AHPO followed by HPLC–diode array detection (HPLC–DAD) analysis [60]. Affenzeller et al. (2019) reported a method for the purification of oxidation products using SPE followed by selective identification using LC–MS [61]. They reported that the introduction of SPE after alkaline oxidation significantly reduced the background signal while maintaining > 70% recovery. Their HPLC–UV–MS method enabled reliable peak identification via accurate mass information of the corresponding UV signals used for quantification. This highly sensitive method could also simultaneously detect EM and PM markers in complex biological samples such as hairs, feathers, and shells [62].

### 2.2. DA-Derived Melanins

Human and primate dopaminergic nigrostriatal neurons produce NM, a dark brown pigment similar to EM and PM in peripheral tissues. NM is a complex polymeric compound present in the central nervous system of humans and other phylogenetically relatively close mammals, including rats, chimpanzees, gibbons, baboons, and more distant ones, such as horses and sheep [63]. NM is composed of not only melanin, but also proteins, lipids, and metal ions [6,8,64,65,66,67]. NM is associated with the pathogenesis of neurodegenerative diseases such as Parkinson’s disease (PD) [68,69,70], and NM is markedly decreased in the SN of PD brains [71,72]. The substantia nigra (SN) and locus coeruleus (LC) are the regions of the brain with the highest concentration of NM [72]. The synthesis of peripheral melanins is mediated by tyrosinase, but the biosynthesis pathway of NM in the human brain has been controversial because the presence of tyrosinase in catecholamine neurons in the SN and LC has been elusive [67,73,74,75,76,77]. A Swedish group (1985) first reported that 5-*S*-cysteinyldopamine (5-*S*-CysDA) was found in the human SN, suggesting the possible incorporation of 5-*S*-CysDA into NM [78]. Then, the same group reported that NM consists of roughly similar amounts of CysDA-derived and DA-derived units [79,80].

Wakamatsu et al. performed chemical analyses to elucidate the structure of pigments in NM isolated from the SN [6,65,81]. AHPO of NM followed by HPLC with UV detection afforded the four degradation products PDCA, PTCA, TDCA, and TTCA. Reductive HI hydrolysis of NM followed by HPLC with electrochemical detection yielded 4-amino-3-hydroxyphenylethylamine (4-AHPEA) and 3-amino-4-hydroxyphenylethylamine (3-AHPEA) as markers of CysDA-derived melanin [7,65,81]. To mimic the processes of NM synthesis and its aging, Wakamatsu et al. (2012) performed model experiments starting with tyrosinase oxidation of various ratios of DA and Cys followed by heating the synthetic NMs. Time course of the NM synthesis and its alteration by heating was followed by chemical degradation methods. Natural NM was found to correspond to a synthetic melanin prepared from a 2:1 molar ratio of DA and Cys followed by heating at 37 °C for 40 days [65]. Furthermore, various catechol metabolites were shown to be incorporated into NM in SN DA and LC NE neurons. The following metabolites are formed by oxidative deamination of catecholamines by monoamine oxidase and the reduction and oxidation by aldehyde dehydrogenase and aldehyde reductase: 3,4-dihydroxyphenylacetic acid, and 3,4-dihydroxyphenylethanol as DA metabolites; 3,4-dihydroxymandelic acid and 3,4-dihydroxyphenylethylene glycol as NE metabolites [65,82,83]. A potential role for NM in the pathogenesis of PD is supported by the epidemiological observations that lightly pigmented individuals and individuals with cutaneous malignant melanoma show higher incidences of PD [84]. Because NM and melanin share many functional features and overlapping biosynthetic pathways, a study by Krainc et al. (2022) hypothesized that genes involved in skin pigmentation and melanogenesis may play a role in the susceptibility of vulnerable midbrain dopaminergic neurons to neurodegeneration [84]. This study also discussed possible mechanisms explaining the association between skin pigmentation and PD [84].

Barek et al. (2018) isolated melanins from several insect samples and studied them using chemical degradation and HPLC analysis of melanin markers [10]. Quantification of various melanin markers revealed that insect melanin differs significantly from mammalian epidermal melanin. Insect EM is primarily made from DHI which appears to be derived from DA. Furthermore, CysDA, but not CD, is the major precursor of insect PM as can be concluded from the very high ratio of 4-AHPEA/4-AHP. This study showed for the first time that insects predominantly use DA as the major precursor for melanin biosynthesis in the cuticle [10]. It also confirmed that PM in many insects is primarily derived from DA rather than from DOPA [10].

Certain bacteria and fungi also produce an insoluble, dark brown, melanin-like pigment. Melanin in *Cryptococcus neoformans*, which is produced by the enzyme laccase, has been thought to be a major virulence factor [85], and the intermediate dopaminechrome involved in EM formation can be produced by the purified laccase [86]. *C. neoformans* is unusual among melanotic fungi in that it requires an exogenous supply of precursor to synthesize melanin. In the melanin of *C. neoformans* cells grown on DA or DOPA, typical degradation markers for DA- and DOPA-derived melanins were identified after chemical degradation with acidic KMnO_4_ oxidation, AHPO, or HI hydrolysis. While PTCA and PDCA were detected, AHP was not found in significant quantities from the cryptococcal pigment [87]. Detectable amounts of AHPEA were found in cells grown on DA, but not on DOPA. These data provide direct chemical evidence for the formation of EM polymers by catecholamine oxidation by laccase alone followed by oxidative coupling of dihydroxyindoles [87].

### 2.3. Usefulness of Melanin Markers in Evaluating Photodegradation of EM and PM

Exposure to excess ultraviolet (UV)A radiation induces degradation/modification of both EM and PM, which is detrimental to pigmented tissues. The UV-induced degradation/modification of melanin structures was evaluated using our chemical degradation followed by HPLC methods [38,88,89]. The photoaging of EM yielded free PTCA (generated by in situ peroxidation) and PTeCA (generated by cross-linking), indicated by the increases of both free/total PTCA and PTeCA/total PTCA ratios [89]. The PTeCA/PTCA ratio has been proposed as a good indicator of EM aging [50]. PM also undergoes degradation/modification upon UVA irradiation. We found that the TTCA/4-AHP and the 4-AHP/3-AHP ratios can serve as good markers for photo-induced oxidative degradation of PM [89,90,91]. The increase in the TTCA/4-AHP ratio reflects a degradative conversion of BT to BZ moieties, whereas the decrease in the 4-AHP/3-AHP ratio may indicate the extent of photodegradation of BT–PM. The increase in the TTCA/PTCA ratio may reflect a degradation of both PM (increase in TTCA due to the conversion of BT to BZ moiety) and EM (decrease in PTCA). Analysis of these markers and their ratios indicates that both EM and PM in human retinal pigment epithelium melanosomes undergo extensive structural changes upon exposure to blue light over a lifetime [90]. Singlet oxygen, in addition to superoxide anions, is photogenerated and quenched by UVA irradiation of EM and PM [38,92,93]. We recently examined the photobleaching of PM having BT and BZ moieties [94,95]. The results showed that synthetically prepared BZ–PM exhibits a higher efficiency to photogenerate singlet oxygen than the synthetic BT–PM, and that 7-(2-amino-2-carboxyethyl)-5-hydroxy-dihydro-1,4-benzothiazine-3-carboxylic acid (DHBTCA) and 6-(2-amino-2-carboxyethyl)-4-hydroxybenzothiazole (BZ-AA) as BT and BZ monomers photogenerate superoxide anions, singlet oxygen, and H_2_O_2_. It was also shown that redox reactions may occur mainly via singlet oxygen and H_2_O_2_ in BZ–AA and via superoxide anions in DHBTCA. The above results indicate that UVA enhanced the pro-oxidant activity of PM, in particular of BZ–PM.

The biological effects of visible light on skin and eyes have attracted increasing attention in recent years [88]. Degradation/alteration of melanin structure induced by the combination of UVA and visible light can lead to decreased photoprotective/cytoprotective capacity of the skin and eyes due to the loss of melanin moieties responsible for their protection. It is also important that the observed changes in melanin structure involve the generation of reactive oxygen species, particularly superoxide anions and singlet oxygen.

We had an opportunity to characterize the phenotypes and examine the effects of sun exposure on the color and structure of EM and PM in alpaca fibers [96]. We analyzed absorbances at 500 nm (A500) and 650 nm (A650), free and total PTCA, and PTeCA as degradative products from EM, and 4-AHP, 3-AHP, and TTCA as degradative products from PM. The ratio of PTeCA/total PTCA increased significantly from the base to the tip in both colors of alpaca fibers, while the ratios of A650/A500 and 4-AHP/3-AHP decreased significantly. These results show that structures made of both EM and PM in alpaca fibers are modified significantly by one year of color change induced by sun exposure. The ratios of A650/A500, PTeCA/total PTCA, and 4-AHP/3-AHP are highly sensitive markers for color change and photodegradation of EM and PM.

Rodríguez-Martínez et al. (2020) examined the influence of endogenous free radical production in the feathers of zebra finches exposed to diquat dibromide in drinking water during a light–dark daily cycle (13L:11D) [97]. As a consequence, the ratio of 4-AHP/TTCA in the flank feathers of birds treated with diquat was lower than that of control birds, indicating a higher content of BZ in PM synthesized by birds treated with diquat. Their results confirmed that free radicals endogenously generated by an organism can induce a structural change in PM, promoting the conversion of BT to BZ.

### 2.4. Spectrophotometric Analysis of Melanins

Melanin is known to be difficult to analyze because it is insoluble in most solvents. Therefore, Ozeki et al. (1995) tried to develop methods to analyze EM and PM separately by solubilizing melanin [98]: (i) for the spectrophotometric determination of EM by solubilizing in hot NaOH in the presence of H_2_O_2_ followed by measurement for absorbances at 350 nm; (ii) for the spectrophotometric determination of PM by mixing vigorously with 0.1 M sodium phosphate buffer, pH 10.5, followed by measurement for absorbances at 400 nm of supernatants. PM can be extracted with alkali from yellow mouse hair and from reddish chicken feathers [40]. However, it was shown that brown-type EM can also be extracted from hairs and feathers with alkaline buffers [99]. Ozeki et al. (1995) showed that PM and brown-type EM can be differentially solubilized under conditions in which black-type EM remains insoluble, and have developed methods to differentially solubilize PM and brown-type EM [98]. For cultured melanocytes, solubilization in hot NaOH or KOH has routinely been used. It should be noted, however, that EM in black mouse hairs cannot be solubilized with this method but can be completely solubilized with the hot-Soluene-350-plus-water method described below.

A spectrophotometric method was developed to measure TM content, estimated by absorbance at 500 nm (A500). This was performed by solubilizing the samples with hot Soluene-350 plus water [100]. Figure 5 shows that the levels of combined amounts of EM and PM in human hairs and epidermis correlated well with the TM level measured spectrophotometrically using this method. This method also provides an estimate of the ratio of EM to PM by analyzing the ratio of absorbance values at 650 nm to 500 nm. The ratio of EM to PM was within the following ranges for different hair colors: 0.12 to 0.14 for red, 0.15 to 0.21 for blonde, 0.19 to 0.26 for light brown, and 0.30 to 0.32 for brown to black [100]. Therefore, the absorbance ratio A650/A500 is useful for estimating the relative proportion of EM to PM. The ratio also reflects the DHICA percentage in EM [101], photodegradation of EM [96], and peroxidative degradation of DA–EM [102]. As an example of the usefulness of the A650/A500 ratio, we have recently shown that brown alpaca fibers show a ratio of 0.265 in the fiber base which decreases significantly to 0.248 in the fiber tip, the decrease being ascribed to solar radiation [96]. The Soluene-350-solubilization-plus-spectrophotometry method is quantitative, precise, and reproducible, although not as specific and sensitive as the chemical-degradation-plus-HPLC method. Therefore, we have been using this method to supplement the HPLC method in phenotyping various tissue samples.

## 3. Various Methods Used for Quantification, Imaging, and Structural Characterization of Melanin

Our method of chemical degradation followed by HPLC for melanin characterization has certain demerits, as it is an invasive and destructive method.

In this section, we describe various recently developed approaches which do not require destruction of tissue samples for melanin characterization. Table 3 summarizes the principles, advantages, and limitations of different analytical methods other than the chemical degradation methods (Table 1).

### 3.1. Electron Paramagnetic Resonance (EPR) Spectroscopy

Electron paramagnetic resonance (EPR) spectroscopy has been used to characterize melanin pigments. In 1982, Sealy et al. reported that EM contains semiquinone with oxygen-centered free radicals, whereas PM contains semiquinonimine type with nitrogen-centered radicals [103,104]. Unlike the singlet EPR spectrum of EM, the PM spectrum contains at least a triplet spectrum but the *g*-factor of the main central peak is practically the same as that of the EM singlet. EPR methods are suitable for analyzing a large number of samples in a short time but are more indirect than the chemical degradative methods. Vsevolodov et al. (1991) reported that both HPLC and EPR methods fit well with the relative values of EM and PM contents in hair samples from newborn lambs (Figure 6) [105]. The HPLC method appears to be more sensitive for detecting low concentrations of PM, while the EPR method is more suitable for mass selection purposes.

Because the EPR signals of EM and PM are different, it has been reported that this method can distinguish between both types of melanin in hair samples and pigmented malignant melanoma in skin [106]. However, due to technical limitations, this method cannot be applied to ex vivo samples and is mainly used for research in the field of melanoma. Pukalski et al. (2020) detected PM-like pigment using EPR spectroscopy [107]. Chikvaidze et al. (2014) reported EPR with free radicals observed in human red hair [108]. They observed that red-hair EPR signals result from a superposition of two spectral shapes, a singlet spectrum as a result of the existence of EM and a triplet spectrum as a result of the existence of PM. The chemical degradation coupled with EPR spectra revealed the presence of EM in sturgeon caviar [109]. EPR spectral features and parameters were in close agreement with those reported for a typical natural EM such as sepia melanin from squid ink [109].

Zadlo et al. (2019) emphasized that the concentration of 4-AHP determined in non-irradiated 5SCD-derived melanin (5SCD-M) and in the photolyzed melanin correlates very well with the intensity of the low-field component of the nitrogen hyperfine line in the observable EPR spectra [94] (Figure 7). The correlation confirms that the nitrogen hyperfine splitting is mostly due to the presence of BT units. It also suggested that their content in the examined PM could be determined by measuring the intensity of the corresponding EPR signal.

### 3.2. ^1^H- and ^13^C-NMR Spectrometry

In order to confirm the molecular structure of extracted melanin, proton and carbon nuclear magnetic resonance (^1^H-NMR, and ^13^C-NMR) analysis can be carried out. Melanin is generally dissolved in a deuterated solvent such as deuterium oxide/sodium deuteroxide (D_2_O/NaOD) [110,111] or dimethylsulfoxide-D6 (DMSO-D6) [112,113]. There are several chemical shifts in the ^1^H and ^13^C-NMR spectra of melanin that can support the correct identification of melanin molecular structures [44]. As an example, characteristic ^1^H-NMR and ^13^C-NMR spectra were obtained for the intracellular melanin extracted from *Lachnum singerianum* YM296 mycelium [114]. In the ^1^H-NMR spectra, the chemical shifts between 6 and 8 ppm were attributed to the aromatic hydrogens of indole and/or pyrrole rings. The observed peaks between 3.2 and 4.2 ppm were caused by the methyl or methylene groups attached to nitrogen and/or oxygen atoms. The presence of an NH-group linked to indole was attributed to the signals within the range of 1.3–2.5 ppm. In the ^13^C-NMR spectra, the peaks between 160 and 185 ppm were typically ascribed to the carbon atoms in carboxyl groups or the carbonyl groups of peptidic bonds [111]. The peaks between 120 and 140 ppm could be generated by aromatic carbons, probably involved in indole or pyrrole systems. The resonance peaks of carbon atoms linked to a nitrogen or sulfur were visible within 50–60 ppm, while the peaks of carbon atoms from methyl and methylene groups were detected within 10–40 ppm [114]. The ^1^H and ^13^C-NMR resonance peaks observed for several melanins have been reported [44,49,111].

Melanins are usually insoluble in solvents, but solid-state NMR overcomes this problem. Recently, solid-state NMR was applied to identify the structural units of melanin in the cell wall of the fungal pathogen *Cryptococcus neoformans* [115,116]. Thus, NMR spectrometry is useful for characterizing structural features (such as a relative ratio of functional groups) of isolated melanins including fossil specimens [49,117]. However, it is not suited for the quantitative analysis of melanins in tissue samples. Additionally, an application of NMR spectrometry to characterize PM is not available. To verify that the organic functional groups present at the surface of the samples also pervade the bulk of the pigmented fossils of sac of Jurassic cephalopods, Glass et al. (2012, 2013) employed ^13^C cross-polarization magic angle spinning solid-state NMR spectroscopy [49,117].

### 3.3. Fourier Transform Infrared Spectroscopy

In preparation for Infrared (IR) spectrum measurement, a portion of the sample is compressed into a KBr pellet. Although the IR spectra differ only slightly regardless of the type of melanin and the extraction procedure, there are some characteristic absorption bands that can be traced to identify the major functional groups of the melanin macromolecules [44,118,119,120].

Fourier transform infrared spectroscopy (FTIR) is a characterization technique, which is rapid, non-destructive, and requires only small-sized samples. As an example of its application, Glass et al. (2012, 2013) reported that FTIR spectra for fossil specimens exhibit diminished absorption bands at 3400 cm^−1^ (the stretching mode of the OH bond), 1605 cm^−1^ (the carbonyl stretch of indole quinone), and 1371 cm^−1^ (in-plane bending modes of OH and NH bonds combined with various modes of aromatic rings) compared to *Sepia officinalis* [49,117]. The overall FTIR spectral features of modern *Sepia officinalis* melanin are also observed in fossil ink sacs. A shoulder at 1710 cm^−1^ is present in all of the fossil ink sac spectra, suggesting a decrease in the preservation of indole quinone units relative to EM from modern *Sepia officinalis* ink [49,117].

### 3.4. Electron Microscopy

Scanning electron microscopy (SEM) is a powerful method for investigating the morphology and particle size distribution of various types of melanin [49,117,121,122,123]. In the literature, the preparation methods for several samples consider sample size, shape, condition, and conductive properties in fungal, bacterial, and other samples (human hair, sepia ink, and retinal pigment epithelium (RPE)) [44]. Transmission electron microscopy (TEM) can examine the fine cellular structures with a high resolution of ~20 nm [124]. SEM and TEM techniques can characterize the melanin samples as they are suitable techniques for evaluating the size and shape of melanin particles. Which technique a researcher chooses often depends on the availability, versatility, and cost of the instruments. Generally, there are many SEM installed all over the world, while the disadvantage of TEM is that the instrument costs are quite high.

### 3.5. Mass Spectrometry

Mass spectrometry (MS) is used to characterize natural and synthetic melanins using different ionization modes such as electrospray ionization (ESI), electron impact (EI), fast atom bombardment (FAB), and matrix-assisted laser desorption/ionization (MALDI), secondary ion mass spectrometry (SIMS) or laser desorption synchrotron positional ionization (synchrotron-LDPI) mass spectrometry. Considering the polymeric morphology of melanin [115] and the different units of melanin, the reported data can be considered rather heterogeneous. Several studies of fungal melanin using ESI have been published [125,126,127]. A recent paper by Yacout et al. (2019) described ESI-MS experiments performed on peroxide-treated synthetic melanin [128]. The spectrum obtained for degraded synthetic melanin samples suggests a partial oxidation of DHICA (*m*/*z* 194 for [M + H]^+^), which has an *m*/*z* of 210 as the most abundant ion. The MALDI-MS technology is widely used for the structural analysis of synthetic and natural melanins. For native melanin samples, MALDI was used not only to characterize the polymers, but also to detect the monomer units of melanin and identify the melanin type. In most cases, MALDI was the method of choice for characterizing the melanin oligomerization process, and most studies that characterized natural melanins used synthetic melanin as a reference material. MALDI-MS was used to study melanogenesis of different types of synthetic melanins synthesized from tyrosine [129], serotonin [130], dopamine [131], DHICA [132] and DHI [133,134], 5,6-dihydroxytryptamine [135], and DOPA [136]. MALDI-MS has also been applied for the analysis of melanin pigments in natural RPE samples [128] and EM in irides [137].

Infrared matrix-assisted laser desorption electrospray ionization (IR-MALDESI) was used as the method of choice for determining the relative melanin content of human hair samples [138]. Unlike MALDI, IR-MALDESI does not require an organic matrix to facilitate the desorption and ionization of target analytes. This technique was used to analyze PTCA content as a quantitative marker of EM. The complexity of melanin analysis stemming from its macromolecular nature and poor solubility in most solvents is driving MALDI as a general approach that offers a way around these challenges. However, considering that the peak separation does not occur during analysis, MALDI can only offer a snapshot of the sample, providing only the molecular ions of the species present in it. Therefore, this method is less suited for non-targeted approaches.

During the last decade, AHPO has increasingly been used to detect and quantify EM traces in fossil samples [49,54,117,139,140]. EM may have maintained its integrity in the fossil record for at least 200 million years [49], whereas most other biomolecules have lost their original properties and polymerized into long chains of hydrocarbons within much shorter periods [141]. Recent studies employ extensive chemical analyses to show that melanin was present in ancient organisms [49,142,143,144,145,146,147,148]. Time-of-flight secondary ion mass spectrometry (ToF-SIMS) is a surface analysis technique that has been extensively employed to investigate the organic content of diverse samples, including pigments in historic paintings [149], and has shown promise as a technique for identifying melanin biomarkers in fossils [148]. Jarenmark et al. (2021) compared the AHPO and ToF-SIMS methods with regard to capabilities in detecting and characterizing EM [150]. According to their results, both AHPO and ToF-SIMS are capable of detecting the different types of EM and monitoring molecular changes caused by artificial maturation at elevated temperatures and pressures, thereby making both methods suitable for the analysis of EM residues in multimillion-year-old fossils. ToF-SIMS provides spatially resolved molecular information that renders it possible to link molecular EM signals to microstructures on a sample surface, whereas AHPO is capable of verifying and quantifying the EM content in a fossil. Thus, a combination of these techniques allows reliable identification and characterization of EM traces in fossilized remains of organisms.

The elemental composition of melanin granules and other components of the hair shaft was determined using multi-isotope imaging mass spectrometry (MIMS) which is a method with unique advantages for the visualization and quantification of stable isotopes and elemental compositions in the study of fine structures of biological samples. MIMS is based on secondary ion mass spectrometry [151]. This method used secondary ions generated from naturally occurring ^16^O, ^12^C, ^14^N, ^32^S, and ^34^S to map and quantify the chemical composition of hair cross-sections with a maximum lateral resolution of 35 nm. Mammalian hair has a highly characteristic elemental composition of many components, including a high sulfur content in keratin-related proteins, making it particularly accessible for studies using this method [152].

### 3.6. Pyrolysis

Due to their polymeric structure, melanin pigments have been analyzed using a combination of pyrolysis gas chr18atography and mass spectrometric detection (Py-GC–MS) [49,120,153,154,155,156]. Synthetic EM models from DOPA, DA, and L-tyrosine [153], and synthetic NMs from 5-*S*-Cys-DA and DA at various molar ratios [155] were subjected to Py-GC–MS. Py-GC–MS experiments identified BT, BZ, and thiazoloisoquinoline derivatives as potential biomarkers for 5-*S*-CysDA-derived PM.

Echinacea is a widely used plant medicine. The melanin from *Echinacea purpurea* was pyrolyzed and the resulting products were separated using gas chromatography and identified using a triple quadrupole mass spectrometer operating in full-scan and multiple-reaction monitoring modes [157]. This result suggested that *E. purpurea* produces three structurally different melanin pigments: EM, PM, and allomelanin. Dzierżęga-Lęcznar described the detection and quantification of PM using a pyrolysis-gas chromatography/tandem mass spectrometry system [158].

The high abundance of indole and methylindole, and the lack of *S*-containing pheomelanin markers in pyrolysates of the analyzed samples suggest that both extant and fossilized ink samples are composed of eumelanin [49,117]. The distribution of major pyrolysis products of the ink sac of the Jurassic cephalopod is similar to that of the modern cuttlefish *Sepia officinalis*. The major pyrolysis products of melanin standard, including pyridine, pyrrole, benzene, phenol, indole, their alkylated homologs, 2-phenylacetonitrile, and 3-phenylpropanenitrile are present in fossil ink sacs. Thiophene and its alkylated derivatives, which are abundantly present in the pyrolysates of fossil ink sac samples, were absent or present below the detection limit in the pyrolysates of ink sacs of extant *Sepia officinalis.*

### 3.7. Raman Spectroscopy

In recent years, Raman spectroscopy has arisen as a powerful tool for the non-invasive analysis of melanins in the tissues of different organisms [159,160,161,162]. Melanin exhibits a characteristic Raman spectrum. Multiple bands in the 1000–1800 cm^−1^ range may contribute to the formation of the Raman spectrum of melanin. Within that wavelength range, the 1220 cm^−1^ band corresponds to the phenolic C–OH stretching vibration and the carboxylic acid C–O stretching vibration, and the 1340 cm^−1^ and 1390 cm^−1^ bands correspond to the indole C–N stretching vibration, and to bands produced by stretching and C=C vibrations of aromatic structures, respectively. Two bands at 1562 cm^−1^ and 1598 cm^−1^ also contribute to the observed Raman spectra, which can be attributed to the stretching mode of the *sp*^2^-hybridized carbon in C=C and the E_2g_ mode of the C–C vibration of the aromatic ring of the indole structure of EM. Due to the superposition of the bands, only two broad bands with maxima of ≈1380 cm^−1^ and ≈1570 cm^−1^ and widths of ≈200 cm^−1^ and 150 cm^−1^, respectively, are observed. Depending on the excitation wavelength, the structural arrangement of melanin, and its biochemical milieu, the amplitude and position of these bands change, allowing the characterization of melanin in vitro and quantification of melanin in vivo [163].

Galván et al. (2013) proposed the use of Raman spectroscopy as a simple, non-invasive technique to identify and quantify melanin in feathers and hair [160]. They used confocal Raman microscopy to analyze synthetic and natural melanin in bird feathers of different species by comparing different excitation wavelengths [161]. Their analysis showed that only laser excitation wavelengths below 1064 nm are useful for the analysis of melanins using Raman spectroscopy, and only the 780 nm laser is useful for feather melanin. These findings demonstrate the ability of Raman spectroscopy to distinguish different chemical forms of melanin, depending on laser power and integration time. As a result, Raman spectroscopy should be applied after preliminary analysis using a range of these parameters, especially when studying fragile biological tissues such as feathers. Galván et al. (2013) found that there are strong correlations between the spectral Raman characteristics (a partial least squares regression (PLSR) model) and 4-AHP and PTCA levels, which indicates that the Raman spectra of melanins can be used to determine their content [160] (Figure 8). Various studies found consistencies in the shape of the Raman spectra of both natural and synthetic pigments, vertebrate and invertebrate melanins, and melanins from different tissues [160,162,164,165,166,167,168]. Therefore, Raman spectroscopy has the potential to become a universal tool for analyzing the hitherto elusive melanins.

Malignant melanoma (MM) is the most aggressive form of skin cancer, approximately 30% of which can arise from pre-existing dysplastic nevus (DN). The diagnosis of DN is now a relevant clinical challenge, as these are intermediate lesions between benign and malignant tumors. Ruiz et al. (2022) evaluated the accuracy of Raman spectroscopy with multivariate analysis (MA) to classify 44 biopsies of MM, DN, and compound nevus (CN) tumors [169]. They first implemented a novel methodology to non-invasively quantify and localize the EM pigment, suspected to be a potential tumor biomarker, by means of Raman spectroscopy imaging coupled with the multivariate curve resolution-alternative least squares algorithm. This represents an improvement over the currently established melanin analysis using HPLC, which is invasive and does not provide information on the spatial distribution of the molecules. Ruiz et al. (2022) showed that the DHI to DHICA ratio is higher in DN than in MM and CN lesions [169]. These differences in chemical composition are used by the partial least squares discriminant analysis algorithm to identify DN lesions in an efficient, non-invasive, fast, objective, and cost-effective method with a high sensitivity and specificity.

### 3.8. Synchrotron Rapid Scanning X-ray Fluorescence

Recently, synchrotron rapid scanning X-ray fluorescence has been successfully developed to fully map trace element distributions in large specimens. Synchrotron X-ray technology has been utilized to map metals chelated by EM. This method may serve as a potential surrogate for pigment distribution in feathers [148] and revealed that the elemental composition of melanosomes varies from organ to organ [123,170,171,172]. A mid-infrared free-electron laser (MIR-FEL) is a synchrotron-radiation-based femto- to picosecond pulse laser. It has unique characteristics such as variable wavelengths in the infrared region and an intense pulse energy. MIR-FEL was applied to analyze the internal structure of fossilized inks from cephalopods for the first time [173].

### 3.9. Near-Infrared Excited Fluorescence

Histochemical staining is a standard tool for visualizing melanin in the skin. However, this method is prone to false-positive results due to melanin dust in the stratum corneum [174,175]. Immunohistochemistry allows for specific labeling of melanogenic and melanosome-specific proteins, but these methods are also highly labile. Recently, optical techniques have emerged as new useful tools for non-invasive skin diagnostics. The unique optical properties of melanin make it a perfect target for ex vivo and in vivo imaging [163]. Unlike other endogenous chromophores in the skin, melanin exhibits a specific broadband absorption that decreases exponentially from short wavelength (UV) to long wavelength (visible light). The long-wave absorption is accompanied by a discernible red/near-infrared (NIR)-excited fluorescence, and melanin is believed to be the major source of NIR fluorescence in the skin. Thus, quantification of skin melanin could be performed using single-photon NIR-exited fluorescence imaging [176,177,178,179,180].

Hani et al. (2014) described the determination of melanin types and relative concentrations by using a non-invasive inverse skin-reflectance analysis [181]. Additional contrast in melanin can be achieved by using two-photon excited fluorescence lifetime imaging microscopy (FLIM), as melanin exhibits a characteristic fast (< 0.2 ns) fluorescence decay [182,183,184,185,186]. Pena et al. (2023) performed multiphoton FLIM analyses of native DHI, DHICA, Dopa–EM, Dopa–PM, mixed EM/PM polymers, and UVA-modified synthetic melanins [187]. They demonstrated that multiphoton FLIM phasor parameters and bi-exponential analyses show promise for characterizing human skin mixed melanins under UVA or other sunlight exposure conditions in vivo.

Other optical techniques such as reflectance confocal laser scanning microscopy [188,189,190] and optical coherence tomography [191,192] are known to be useful in detecting pathological melanocytic lesions. Photoacoustic signals can also be used to quantify melanin in the skin by measuring long wavelength absorption tails [193,194,195].

### 3.10. Fate Tracing of [U-^13^C]L-Tyrosine Using LC–MS

Chen et al. (2021) developed a novel method to measure both EM and PM synthesis by tracing the fate of [U-^13^C]L-tyrosine using liquid chromatography–mass spectrometry [196]. Using this method, they confirmed a previous report [197] regarding the differences in melanin synthesis between melanocytes derived from individuals with different skin colors and MC1R genotypes and could apply this method to quantify de novo synthesis of both EM and PM over unprecedently brief time periods since the carbon atoms of L-tyrosine are retained through all the steps of EM and PM synthesis. They also revealed that a distinct mechanism that specifically changes the pH of melanosomes induces the synthesis of new EM and PM. Since L-tyrosine fate tracing is compatible with untargeted liquid chromatography–mass spectrometry-based metabolomics, this approach allows extensive measurements of cellular metabolism in combination with melanin metabolism, can assess multiple functions of melanogenesis, and is expected to shed new light on the mechanism of melanogenesis.

### 3.11. Pump–Probe Method

Pump–probe imaging techniques can be used to distinguish between EM and PM at the dermo–epidermal junction to assess the metastatic potential of melanin lesions and to characterize the packing of melanin oligomers within aggregates [198,199,200,201,202,203,204,205,206,207]. Pump–probe microscopy is a combination of laser scanning microscopy and pump–probe spectroscopy [204]. In the pump–probe method, two beams of light (or particle beams) are used. One of the beams, a laser beam (pump) is used to excite a sample, and a time-delayed laser beam (probe) interacts with the excited sample, thus probing it at various precisely controlled temporal moments. The pump and probe can be from the same source or from two different sources. Many pump–probe methods use optical pulses. This method is commonly used to measure ultrafast phenomena by using short laser pulses. It enables time-resolved measurements of picoseconds or less, which is difficult to follow with electrical methods and is widely used in the field of photochemistry. Using this, the process of chemical reactions can be measured in real-time. A method for establishing a melanin standard based on nonlinear pump–probe microscopy was introduced to form the basis of a biomarker for metastatic melanoma. Pump–probe microscopy can be used to nondestructively examine the chemical and physical properties of melanin in cell cultures and formalin-fixed paraffin-embedded slices of primary melanoma. Over the past few years, advances in nonlinear laser microscopy have provided a new and important approach for diagnosing and grading melanoma by visualizing the molecular details of melanin in pigmented skin lesions [200,204,208]. This is very important because early detection of aggressive cancers is difficult. A problem is that standard diagnostic protocols (excision, staining, pathological evaluation) fail to identify most of the dangerous early lesions.

Recent studies have shown that melanin with femtosecond pump–probe signals is heterogeneous in skin, conjunctiva, and vulvar tissue, and this heterogeneity is of direct clinical concern. Using a 720 nm pump and 817 nm probe, average pump–probe signatures are observed from negative-dominant short-lived signals in normal moles to positive-dominant long-lived signals in metastatic cancers [198]. It was noted that the difference between the two signals was consistent with the difference between EM and PM. This may be due to known differences in bulk composition between benign moles, dysplastic nevi, and melanomas. Chemical changes in EM observed by NIR pump–probe microscopy in pigmented skin lesions are independent of PM content but are associated with changes in EM self-assembled structures [200]. By relating the pump–probe signal of EM to its structure and function in melanoma progression, pump–probe microscopy is not only a useful imaging tool for melanoma, but also for the structure–function relationship of EM in the human pigmentary system [199].

### 3.12. Hair and Skin Color Parameters—ITA, and Colorimetric Parameters

In this section, we describe good correlations between individual typological angle (ITA)-based classification or the colorimetric parameters (L*, a*, and b*) and the chemical degradation markers.

Sensitivity to sunlight is known to be highly dependent on the degree of constitutive pigmentation of the skin. Del Bino et al. (2018) reported that the ITA-based classification of skin color correlates with constitutive pigmentation and is physiologically relevant in interaction with geographical regions [209]. This may help identify different responses to UVR exposure that correspond to the UVR sensitivity of a particular skin color type, supporting the concept of personalized photoprotection [52,209]. The color of human skin is determined using the TM, EM, and PM combined, and the ratio between the black-to-brown EM and yellow-to-red PM which are distributed through the epidermis [36,210]. Skin-color assessment using the famous Fitzpatrick phototype classification system widely used by dermatologists was originally created for Caucasian skin and is based on self-reported erythema sensitivity and tanning ability, leading to potential limitations regarding quantification and reliability. Additionally, this classification system has been shown to be inappropriate for some populations, including Asians, African Americans, and possibly Hispanics as well as multi-ethnic populations [209,211]. For dermatological and cosmetic research, there is a need to objectively classify the skin color. For this purpose, another classification system based on the determination of the colorimetric parameters L* (lightness axis), a* (red–green axis) and b* (yellow–blue axis) and ITA has been proposed [212]. Pigmentation was measured using the L* parameter, which varies from black (value 0) to white (level 100), and the b* parameter, which varies from yellow (positive value) to blue (negative value). ITA is calculated using the following formula: ITA = [arctan(L* − 50)/b*] × 180/3.14159. Using this classification system, skin tone types are classified into six groups, ranging from very light to dark. The validity of the ITA classification has been evaluated with regard to its correlation with constitutive pigmentation using Fontana–Masson staining [213]. The ITA value also highly correlates with the TM content measured indirectly as an A500 value by spectrophotometry after dissolving epidermal samples in Soluene-350 (R^2^ = 0.83, *p* < 0.0001) [52]. Analysis of a large number of skins showed a good correlation (R^2^ = 0.85, *p* < 0.0001) between ITA and PTCA for EM, and (R^2^ = 0.72, *p* < 0.0001) between ITA and TTCA for PM, but showed a low correlation between ITA and 4-AHP (R^2^ = 0.24, *p* < 0.0001) [52].

Itou et al. (2019) found that Japanese women’s hair color darkens with age due to increases in melanosome size, the amount of melanin, and mol% of DHI units, which have a high absorbance [101]. Next, they extended the same analyses to men’s hair to examine gender differences in hair color, melanin composition, and melanosome morphology [214]. Men’s hair also tends to darken with age, but it was darker than women’s hair at a young age. In men’s hair, there was no age-dependency of DHI mol% as in women’s hair, but melanosome size increased with age and TM content increased as well, suggesting that these findings were correlated with hair color. Age-dependent analyses revealed significant sex differences in melanosome features such as A650/A500, PM mol%, and the minor axis of melanosomes. They suggest that TM content, PM mol%, and DHI mol% are correlated with hair color.

The skin color and skin redness have also been measured by using Antera^(®)^ 3D, Mexameter ^(®)^ and colorimeter ^(®)^ [215] and the method for skin-reflectance reconstruction from camera images has been improved [216]. Masuda et al. (2009) described an innovative method to measure skin pigmentation using a spectrum resolution method [217].

Alaluf et al. (2002) compared human ethical skin color via the correlation between tristimulus (L*, a*, b*) and melanin amount using the solubilization methods of alkaline-soluble and alkali-insoluble melanin by Ozeki et al. (1995) [98,218,219]. In general, it is known that constitutively dark skin types have lower L* values, higher a* values, and higher b* values than constitutively light skin types. Total epidermal melanin content appears to be the primary determinant of L* values in human skin, while melanosome size also has a significant but more subtle influence on L* values. Epidermal melanin content has a strong positive contribution to a* values and is also a major contributor to b* values in lighter skin types. As a result, studies have shown that absolute L*a*b* values in objectively measured human ethical skin color, regardless of ethnicity or photo-exposure, are all substantially determined by the amount of melanin in the epidermis [219].

### 3.13. Tape Stripping

Tape stripping (TS) is widely used in dermatological research as a minimally invasive method of sampling the epidermis, avoiding the need for skin biopsies [220,221,222]. Stratum corneum (SC) is easily accessible and collected using adhesive tape. Given its minimal invasiveness and simplicity, TS is particularly useful in dermatological research when repeated sampling over time is desirable, such as in studies in pediatric populations or clinical trials. The advantages are that (i) TS is a simple and minimally invasive method for collecting SC samples, and (ii) SC samples are suitable for measuring a wide range of biomarkers using different analytical platforms such as proteomics, lipidomics, and transcriptomics. However, as limitations, the kinetics of biomarker translocation from the living epidermis to the SC are largely unknown. To correct for variable amounts of SCs removed using tape, it is often necessary to normalize biomarker expression to SC amounts adhered to the tape.

Matsunaka et al. (2017) established a non-invasive method for quantifying EM and PM content of the SC using TS and HPLC [220]. This non-invasive method can serve as a marker for pathology in skin pigmentation diseases such as malignant tumors.

### 3.14. Elemental Analysis

Elemental analysis, although a classical method, can be used to characterize synthetic melanins and purified natural melanins [1]. Using this approach, melanin can be easily classified into subtypes by evaluating the nature of the elements from the pigment and/or determining specific element ratios (C:H:N:O:S). As noted earlier in this review, EM lacks sulfur in its structure unless subjected to HCl hydrolysis to remove proteins [223]. *Sepia officinalis*, which is pure EM, is used as a reference standard [1]. On the other hand, PM contains a high amount (as much as 10%) of sulfur [224]. Several fungi are able to produce other types of melanin pigments: 1,8-dihydroxynaphthalene (DHN)-melanins which are synthesized via the polyketide pathway, and pyomelanins which are soluble melanins produced through hydroxyphenylpyruvate and homogentisic acid [225,226]. Fungal melanins include DHI and DHICA monomer units with 6–9% nitrogen or 1,8-DHN with no nitrogen in its structure [227].

To gain further insights into the chemistry of fossilized materials, Glass et al. (2012, 2013) used elemental analysis, py-GC–MS, FTIR, and X-ray photoelectron spectroscopy [49,117]. This is a powerful combination of techniques for identifying structural properties of complex organic constituents. Elemental analysis (C, H, and N) of fossil melanin and fossil sediment can also provide a quantitative measure of the C, N, and H present in each sample. Elemental analyses revealed that fossil ink sacs have very similar C, N, and H distributions compared to extant ink sacs, despite differences in the dimensions of their granules. Some of the carbon and hydrogen detected using elemental analysis in fossilized cephalopod ink can be accounted for by the presence of diagenetic phases established by the mineralization of fossil specimens [49,117].

## 4. Conclusions

We here reviewed various methods for melanin characterization using chemical analytical methods (AHPO, HI hydrolysis, and soluene-350 plus water), surface morphology, and structural elucidation as summarized in Figure 9. The analytical tools available for melanin analysis, although mostly complementary, provide the specific information needed to draw accurate conclusions about the melanin sample analyzed. For quantitative and qualitative approaches, the preferential method involves the chemical degradation (AHPO and HI hydrolysis) of the sample followed by HPLC or LC–MS measurement of the degradation products because the structural units of the given melanin can be evaluated quantitatively. Although not as specific and sensitive as the chemical analysis plus HPLC, Soluene-350 solubilization plus spectrophotometry is quantitative, precise, and reproducible. The results of the chemical analysis plus HPLC and the ITA value highly correlated with the TM content measured indirectly as an A500 value using spectrophotometry after dissolving samples in Soluene-350 plus water.

The chemical and quantitative analyses of melanin showed that DHICA–melanin in EM exhibits a potent hydroxyl radical scavenging activity, whereas DHI–melanin does not [228], and that BZ moieties in PM are more photoreactive than BT moieties [94]. If only qualitative information is required, the best analytical methods are those that can classify melanin pigments based on their melanin structural units, such as elemental analysis, Py-GC–MS, ToF-SIMS, Raman, and MALDI. For macromolecular structure elucidation, the best approaches are microscopy techniques (TEM and SEM) along with MALDI, FTIR, Raman, and NMR. Additionally, we should not neglect basic measurements such as colorimetry, solubility, and UV/visible spectra, which can provide valuable information in the shortest amount of time. Very recently, Cao et al. (2023) investigated synthetic routes to a close mimic of natural PM [229]. The route employing 5SCD among the different oxidative polymerization routes was verified as a close analogue of extracted PM from human and birds. Comparison of structures between biomimetic and natural PM was performed using solid state ^13^C-NMR, FTIR, and EPR, and it was reported that synthetic PM closely mimics the structure of natural PM [229]. The ultimate aim of all these experimental methods is that melanin analysis can be used in a wide range of biomedical and technological applications, from skin cosmetics (especially photoprotection) to radioprotection, thermoregulation, and phytopathogenicity.

**Figure 9 ijms-24-08305-f009:**
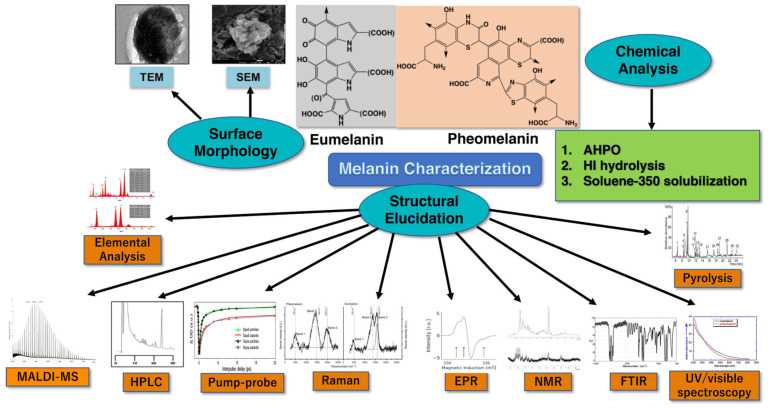
Melanin characterization using AHPO, HI hydrolysis, and soluene-350 solubilization for chemical analysis; TEM and SEM for surface morphology; MALDI-MS, HPLC, Pump–probe, Raman, EPR, NMR, FTIR, Elemental analysis, Pyrolysis, and UV/visible spectroscopy for structural elucidation. Modified with permission from ref. [230]. Copyright 2011, Elsevier.

## Figures and Tables

**Figure 1 ijms-24-08305-f001:**
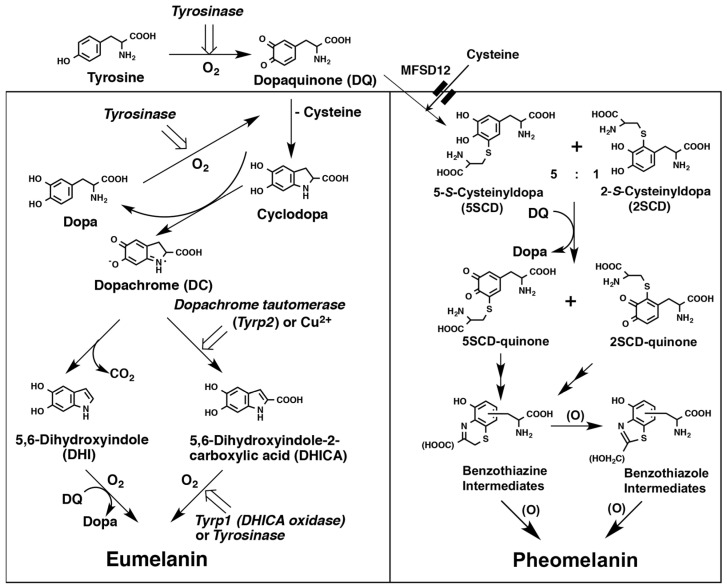
Biosynthetic pathways leading to the production of EM and PM. Note that the activities of tyrosinase, Tyrp1 and Dct/Tyrp2, and the precursor tyrosine are involved in the production of EM, while only tyrosinase (and the precursors tyrosine and cysteine) is necessary for the production of PM. While mouse Tyrp1 acts as a DHICA oxidase as shown in the Figure, the function of human TYRP1 has not been established. In the reaction of DQ and cysteine, 5SCD and 2SCD are produced at a ratio of 5:1, respectively [33]. Benzothiazine and Benzothiazole intermediates are made from 5SCD–quinone and 2SCD–quinone, respectively. MFSD12 (major facilitator superfamily domain-containing protein 12; depicted by double lines) is a component of the melanosomal cysteine import system. Modified from ref. [4].

**Figure 2 ijms-24-08305-f002:**
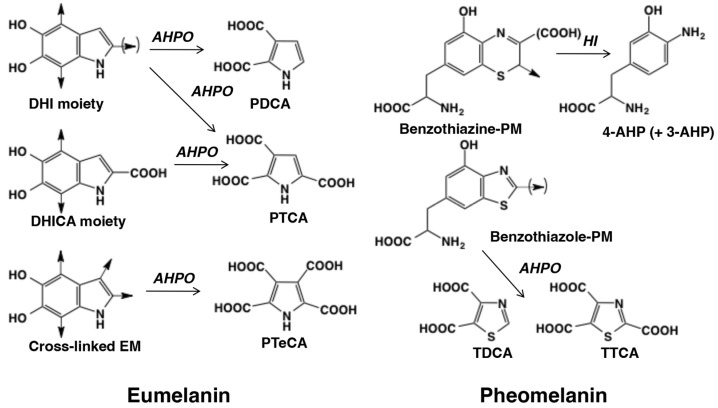
Alkaline hydrogen peroxide oxidation (AHPO) and hydroiodic acid (HI) hydrolysis of eumelanin (EM) and pheomelanin (PM) yielding various melanin markers. 5,6-dihydroxyindole (DHI), 5,6-dihydroxyindole-2-carboxylic acid (DHICA), pyrrole-2,3-dicarboxylic acid (PDCA), pyrrole-2,3,5-tricarboxylic acid (PTCA), pyrrole-2,3,4,5-tetracarboxylic acid (PTeCA), thizole-4,5-dicarboxylic acid (TDCA), thiazole-2,4,5-tricarboxylic acid (TTCA), 4-amino-3-hydroxyphenylalanine (4-AHP), 3-amino-4-hydroxyphenylalanine (3-AHP). Adapted from ref. [48].

**Figure 3 ijms-24-08305-f003:**
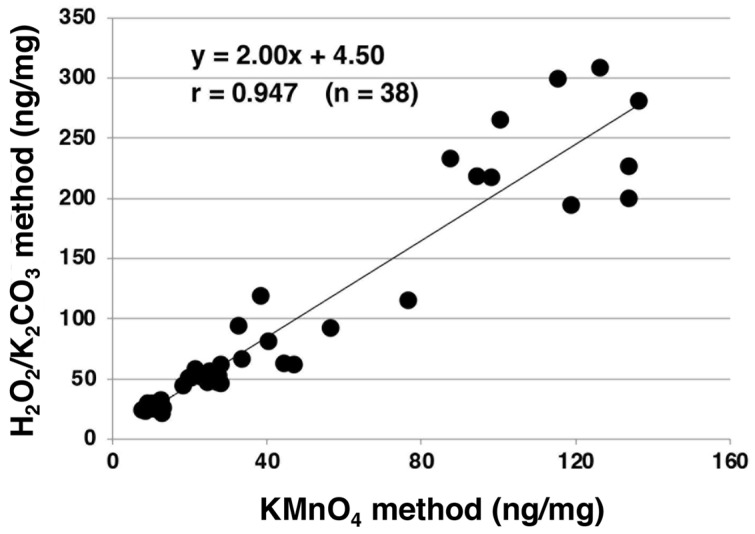
Correlation of the PTCA amounts generated using AHPO and acidic KMnO_4_ oxidation from human hair samples. The correlation was significant (*p* < 0.0001). *n* = 38. Adapted with permission from ref. [43]. Copyright 2011, John Wiley and Sons.

**Figure 4 ijms-24-08305-f004:**
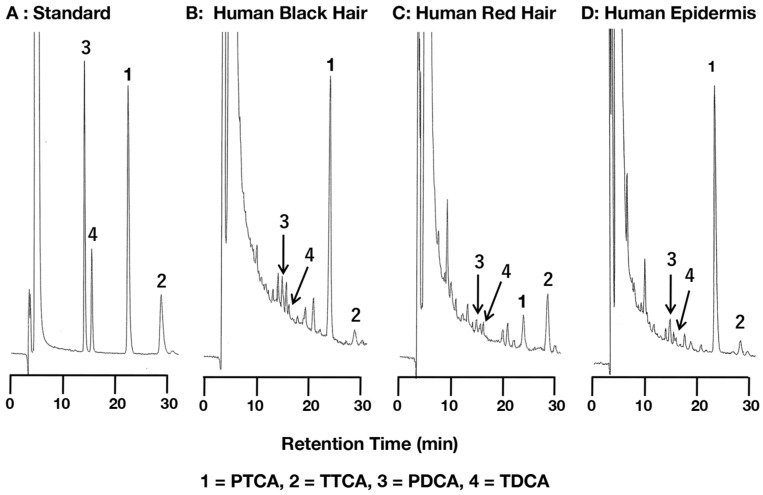
HPLC chromatograms of standard melanin markers (**A**), and AHPO mixtures from human black hair (**B**), from human red hair (**C**), and from dark human epidermis (**D**). Adapted from ref. [48].

**Figure 5 ijms-24-08305-f005:**
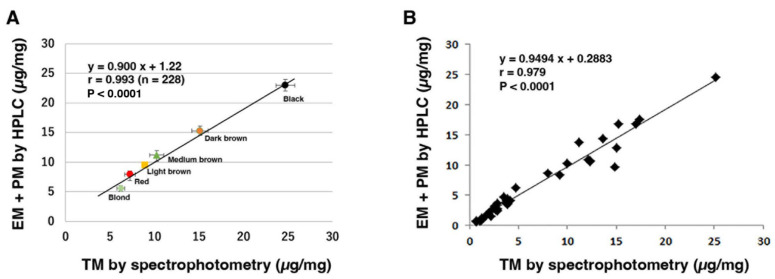
Correlation between TM using a spectrophotometric method and EM + PM using an HPLC method. (**A**) Six different colors of human hair samples (a total of 228 samples). TM was measured at 500 nm. Bars represent SEM. Adapted with permission from ref. [43]. Copyright 2011, John Wiley and Sons. (**B**) A total of 35 human epidermis samples with diverse pigmentation. Adapted with permission from ref. [52]. Copyright 2015, John Wiley and Sons.

**Figure 6 ijms-24-08305-f006:**
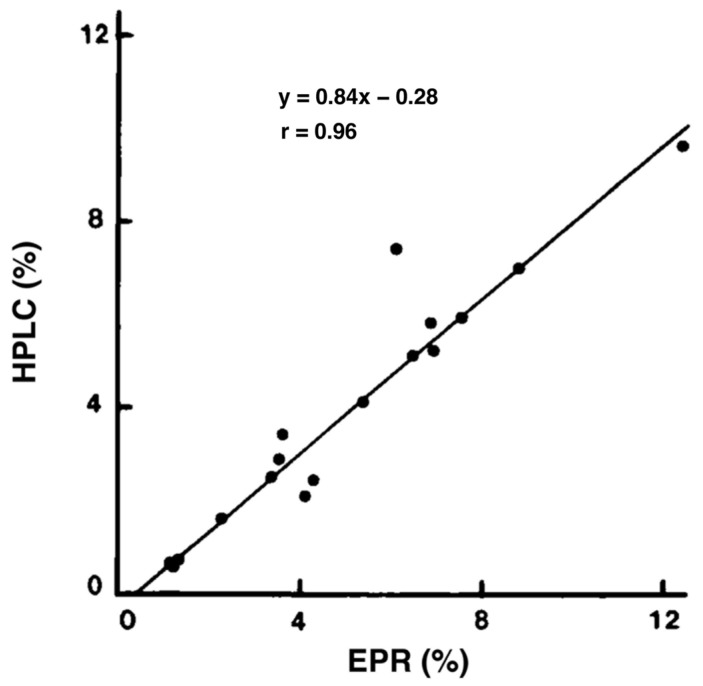
Correlation between EPR and HPLC estimates of TM contents (EM + PM) in hair samples. *n* = 17. Adapted with permission from ref. [105]. Copyright 2006, John Wiley and Sons.

**Figure 7 ijms-24-08305-f007:**
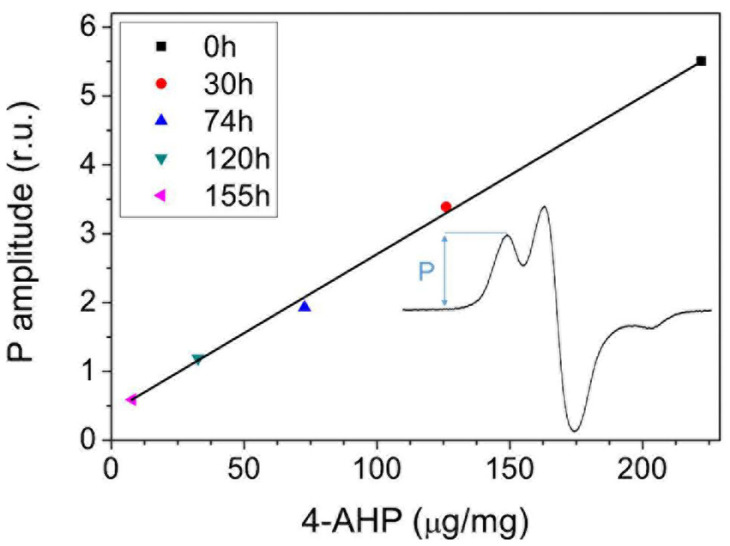
Intensity of the low-field component (P) of the EPR signal of 5SCD-M vs. content of 4-AHP in selected samples subjected to photolysis. Adapted with permission from ref. [94]. Copyright 2018, John Wiley and Sons.

**Figure 8 ijms-24-08305-f008:**
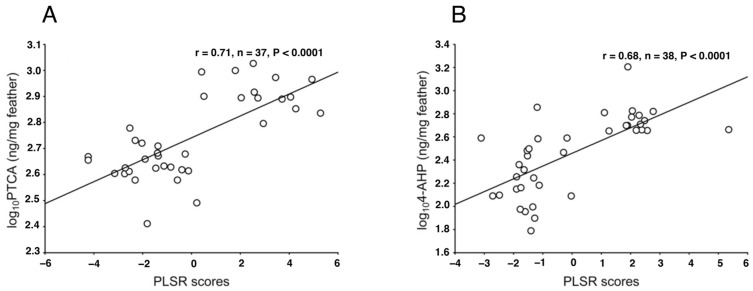
Relationship between EM and PM levels measured using HPLC and the scores of PLSR factors collecting information on characteristics of the Raman spectra. (**A**,**B**) are the feathers of red-legged partridges. Adapted with permission from ref. [160]. Copyright 2013, John Wiley and Sons.

**Table 1 ijms-24-08305-t001:** Comparison of chemical degradative methods to characterize EM and PM.

**Eumelanin (EM)**	**Methods**	**Advantages**	**Limitations**
	Acidic KMnO_4_ oxidation [34,35,36,41]	PTCA is highly specific	(1) Difficult to perform (ether extraction) (2) PDCA cannot be determined
	Alkaline H_2_O_2_ oxidation (AHPO) [43,47,48]	(1) Easy to perform by direct injection (2) PDCA can also be determined	Artificial production of PTCA from PM
**Pheomelanin (PM)**	**Methods**	**Advantages**	**Limitations**
	HI hydrolysis [34,46]	(1) 4-AHP is highly specific (2) Electrochemical detection (ECD) is highly sensitive	Difficult to perform (HI evaporation)
	Alkaline H_2_O_2_ oxidation (AHPO) [43,47,48]	(1) Easy to perform (2) TTCA and TDCA can be used as markers	Lower sensitivity for TTCA and TDCA

**Table 2 ijms-24-08305-t002:** Reproducibility of melanin markers with the improved HPLC method (*n* = 5).

Hair	PTCA	PDCA	PTeCA	TTCA	TDCA
Black	205 ± 4.4 (2.1%)	19.2 ± 0.60 (3.2%)	82.0 ± 3.6 (4.4%)	43.4 ± 0.94 (2.2%)	19.4 ± 1.9 (9.7%)
Red	23.4 ± 1.2 (5.2%)	10.7 ± 0.52 (4.8%)	12.9 ± 1.6 (12.4%)	169 ± 6.7 (4.0%)	26.7 ± 0.86 (3.2%)

Note: Values are means ± SD in ng/mg; CVs are shown in parentheses. Adapted from [48]. Homogenates of human hair were analyzed consecutively for 5 days using AHPO–HPLC.

**Table 3 ijms-24-08305-t003:** Principles of methods, advantages, and limitations of various analytical methods.

Analytical Method	Principles of Methods	Advantages	Limitations
Spectrophotometric analysis	(1) Soluene-350 plus water (2) NaOH	(1) Conventional and inexpensive (2) Solubilizes both EM and PM (Soluene-350) (3) Good correlation between TM using spectrometry and melanin contents using HPLC	(1) Some background of absorbance from tissue constituents such as proteins and the viscosity of the solvent (Soluene-350) (2) Low sensitivity and selectivity
Electron paramagnetic resonance (EPR) spectroscopy	Method for studying materials that have unpaired electrons	(1) Non-invasive and non-destructive (2) Method allows to distinguish between EM and PM (3) Possibility to study physicochemical properties of melanins after binding chemicals and metals	(1) Short lifespan of some radicals (2) Instrument is expensive (3) Sub-units of EM and PM cannot be quantified (4) Unable to distinguish between 1,8-dihydroxynaphthalene (DHN)-, pyo-, and eumelanins
^1^H and ^13^C-NMR spectrometry	The method is a physicochemical technique and based on the physical phenomenon of magnetic resonance	(1) High selectivity (2) Solid samples can be analyzed (3) Useful for characterizing structural features such as the ratios of aliphatic and aromatic ^1^H and ^13^C	(1) Not laboratory-based (2) Instrument is expensive and large (3) Not suited for the quantitative analysis of melanins in tissue samples
Fourier transform infrared spectroscopy (FTIR)	(1) The technique is used to obtain an infrared spectrum of absorption or emission from a solid, liquid, or gas (2) Method based on absorption of infrared radiation and excitation of oscillatory levels, to study the presence of chemical functional groups	(1) Highly sensitive and quick method to achieve a high-quality spectrum (2) Ability to analyze solid, liquid, or gas phase samples (3) Good signal-to-noise ratio (4) Non-destructive (5) Scan within 1–2 s (6) Capability to create chemical distribution images	(1) Only single beam (2) Instrument is expensive (3) The Rayleigh criterion reduces the spatial resolution of chemical images compared to resonance Raman spectroscopy images
Electron microscopy	The method uses a beam of electrons and their wave-like characteristics to magnify an object’s image	(1) Powerful method for investigating the morphology and particle size distribution of various types of melanin	(1) Instrument is expensive and large
Mass Spectrometry (MALDI–MS, ToF–SIMS, GC/MS)	The method is to generate ions from either organic or inorganic compounds, using any suitable method to separate these ions by their mass-to-charge ratio (*m/z*) and to detect them qualitatively and quantitatively by their respective *m/z* and abundance	(1) Highly sensitive and selective (2) Able to identify/quantify components of mixtures (3) Possibility of combination with other techniques such as HPLC (LC–MS) and gas chromatography (GC–MS) (4) Very precise, rapid, and sensitive method (5) Measurability with very small amounts (ppm levels) of samples (6) Able to be used for both qualitative and quantitative analysis of chemicals	(1) Costly, the system needs a skilled technician and is not a portable system (2) Large instrument (3) Not laboratory-based
Pyrolysis	The method is a process by the thermal decomposition occurring in the absence of oxygen or any other oxidants, and one of the most common methods in thermal conversion technology of biomass	(1) Simple and fast process (2) More suitable technology for bio-oil production (3) Scale-up is economically feasible (4) Efficient energy conversion	(1) Possibility of requiring additional energy (2) Increased biochar production (3) Biomass collection is its main problem of industrialization (4) Limited commercial experience (5) Low thermal stability (6) Production of pyrolytic water
Raman spectroscopy	Raman is a scattering technique which is based on Raman effect, i.e., the frequency of a small fraction of scattered radiation is different from the frequency of monochromatic incident radiation. It is based on the inelastic scattering of incident radiation through its interaction with vibrating molecules.	(1) Non-invasive and non-destructive (2) Good correlation between the results obtained from the chemical degradations and the Raman spectrum (3) Laboratory-based and portable instruments (4) Ability to analyze materials in sealed transparent containers (5) Capability to create chemical distribution images	(1) Difficulty to build calibration curves (2) Trouble with fluorescent or strongly absorbing materials (black materials) (3) Chemical species with a high Raman scattering cross-section, even in small concentrations, may give intensive bands covering other, important analytical bands in the spectrum (4) Problematical to obtain an accurate spectrum of amorphous materials (5) Potential for ignition of explosives (6) Possibly long collection times (7) Low sensitivity (8) Instrument is expensive (9) Specific power requirements
Synchrotron rapid scanning X-ray fluorescence	The method provides spatial distribution and quantification of ions in samples ranging in length from mm to submicron	(1) Able to visualize the distribution of ions. (2) Useful to study ionic processes at very small scales. (3) Useful for in vivo analysis that requires room temperatures and pressures or high detection limits of about 1-100 mg/kg or nanoscale resolutions of about 50 nm.	(1) Necessitates very large and sophisticated facilities that are not readily available to the public. (2) Sample preparation necessitates extensive technical knowledge and is delicate. (3) The weight of elements influences the distribution of elements in samples.
Near-infrared excited fluorescence	Light of 800 to 2500 nm, which is said to be in the near-infrared region, irradiates the sample, and other components are simultaneously measured by making full use of statistical methods for the absorbed wavelengths.	(1) High sensitivity, non-invasiveness, and lack of radiation hazard (2) Low background interference (3) High penetration (1–10 mm) (4) Low tissue damage (5) Fast, real-time display, relatively low cost, portability (6) Sample preparation is almost unnecessary	(1) Broad peak shape (2) Quantitative analysis is difficult. (3) Moisture and granularity are influential factors
Pump–probe method	Using two beams of light (or particle beams), one beam (the pump beam) illuminates a material to induce changes in the material, and the other beam (the probe beam) measures those changes	(1) Non-destructive to cells and tissues (2) Imaging of endogenous pigments with three-dimensional spatial resolution (3) Efficient discrimination between hemoglobin and melanin (4) Possible discrimination of melanoma based on EM/PM ratio	(1) Difficulty of accurate detection in samples with weak signals, such as highly diluted solutions (2) Difficulty in suppressing scattered light, especially when the pump and probe spectra overlap (3) Contamination of the signal from the probe beam by a strong pump, when the pump and probe pulses overlap in both space and time at the sample surface (4) Instrument is expensive
ITA and colorimetric parameters	(1) A colorimeter is a light-sensitive tool used to measure the absorption and transmission of light passing through a sample matrix. (2) ITA measures constitutive pigmentation. (3) ITA and colorimetric parameters are facultative.	(1) Inexpensive, fast, and simple operation of the colorimetric method (2) A fast and convenient method compared to the volumetric or gravimetric process (3) Does not require an experienced person to handle the colorimetric method (4) Applied to the quantitative analysis of colored compounds with the colorimetric method (5) Portable system to easily carry for the colorimetric method (6) Low cost	(1) Unable to analyze colorless compounds (2) Needs a high number of samples for analysis (3) Low sensitivity (4) Generation of errors in the results by interference from material of the same color (5) Reflection of light on some surfaces makes measurements difficult
Tape stripping	A method to collect and remove the stratum corneum by attaching adhesive tape such as cellophane tape to the skin	(1) Minimally invasive (2) Simple and easy to perform (3) Easy collection of stratum corneum	(1) Unknown kinetics of biomarker translocation from the living epidermis to stratum corneum
Elemental analysis of C, H, N, and S	Method to determine the elemental composition of the sample	(1) Ability to distinguish EM, PM, DHN-, and pyomelanins (2) Fast (3) Low cost	(1) High purity of the sample required (2) It may be difficult to distinguish between DHN- and pyomelanins.

## Data Availability

Not applicable.

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
