# Peer review of "Recent Advances in Characterization of Melanin Pigments in Biological Samples"

_ijms, 2023, doi:10.3390/ijms24098305_

Round 1

Reviewer 1 Report (Previous Reviewer 2)

The manuscript entitled “Recent Advances in Characterization of Melanin Pigments in Biological Samples” is a resubmission. The authors improved the manuscript compared to the original submission giving more precise information about the discussed data.

Moderate editing of English language is required.

Author Response

  1. Moderate editing of English language is required.

I appreciate your comments. I appreciate your comments. The English editing of manuscript has been proofread by a native speaker who is our collaborator at our university.

<Additional correction>

  1. We corrected Figure 3 to add the vertical and horizontal axis units.
  2. We corrected Figure 5A to add the p value.
  3. Recently, we noticed that a new relevant paper has been published. Thus, we added the following sentence at the end of the conclusion:

"Very recently, Cao et al. (2023) investigated synthetic routes to a close mimic of natural PM. The route employing 5SCD among the different oxidative polymerization routes was verified as a close analogue of extracted PM from human and birds. Comparison of structures between biomimetic and natural PM was performed by solid state 13C-NMR, FTIR and EPR, and it was reported that synthetic PM closely mimics the structure of natural PM [234]."

  1. Reference #160 and #234 were newly added:
  2. #160.  StÄ™pieÅ„, K.; Dzierżęga-LÄ™cznar, A.; Kurkiewicz, S.; Tam, I. Melanin from epidermal human melanocytes: study by pyrolytic GC/MS. J. Am. Soc. Mass Spectrom. 2009, 20, 464-468. https://doi.org/10.1016/j.jasms.2008.11.003
  3. #234.  Cao, W.; Mao, H.; McCallum, N.C.; Zhou, X.; sun, H.; Sharpe, C.; Korpanty, J.; Hu, Z.; Ni, Q.Z.; Burkart, M.D.; et al. Biomimetic pheomelanin to unravel the electronic, molecular and supramolecular structure of the natural product. Chem. Sci. 2023, 14, 4183. https://doi.org/10.1039/d2sc06418a

Reviewer 2 Report (Previous Reviewer 4)

The manuscript was improved and the authors have responded to the suggestions made by all reviewers and the academic editor. I only heve very few remarks:

In Fig 1, what is the meaning of two bold traces in the first arrow, in the right side?

Figures 6 and 7 are superimposed.

In Figure 9, is lacking elemental analysis?

Author Response

The manuscript was improved and the authors have responded to the suggestions made by all reviewers and the academic editor. I only have very few remarks:

  1. In Fig 1, what is the meaning of two bold traces in the first arrow, in the right side?

 This means cysteine transmembrane transporter. I added “depicted by double lines”.

  1. Figures 6 and 7 are superimposed.

I received an email from the Assistant Editor asking me to mention this fact to the Reviewer 2, as this happened when the Assistant Editor did the layout. The layout of Figures 6 and 7 is now improved.

  1. In Figure 9, is lacking elemental analysis?

I appreciate your comments. I added the figure of the elemental analysis. Thus, I remade Figure 9 and Graphical Abstract. I also added a diagram of pyrolysis in Figure 9.

<Additional correction>

  1. We corrected Figure 3 to add the vertical and horizontal axis units.
  2. We corrected Figure 5A to add the p value.
  3. Recently, we noticed that a new relevant paper has been published. Thus, we added the following sentence at the end of the conclusion:

Very recently, Cao et al. (2023) investigated synthetic routes to a close mimic of natural PM. The route employing 5SCD among the different oxidative polymerization routes was verified as a close analogue of extracted PM from human and birds. Comparison of structures between biomimetic and natural PM was performed by solid state 13C-NMR, FTIR and EPR, and it was reported that synthetic PM closely mimics the structure of natural PM [234].

  1. Reference #160 and #234 were newly added:
  2. #160.  StÄ™pieÅ„, K.; Dzierżęga-LÄ™cznar, A.; Kurkiewicz, S.; Tam, I. Melanin from epidermal human melanocytes: study by pyrolytic GC/MS. J. Am. Soc. Mass Spectrom. 2009, 20, 464-468. https://doi.org/10.1016/j.jasms.2008.11.003
  3. #234.  Cao, W.; Mao, H.; McCallum, N.C.; Zhou, X.; sun, H.; Sharpe, C.; Korpanty, J.; Hu, Z.; Ni, Q.Z.; Burkart, M.D.; et al. Biomimetic pheomelanin to unravel the electronic, molecular and supramolecular structure of the natural product. Chem. Sci. 2023, 14, 4183. https://doi.org/10.1039/d2sc06418a

This manuscript is a resubmission of an earlier submission. The following is a list of the peer review reports and author responses from that submission.

Round 1

Reviewer 1 Report

I have been through the review manuscript titled "Recent Advances in Characterization of Melanin Pigments in Biological Samples".

The Keywords should be limited. 

Graphical abstract will be better for illustrating their study of focus.

Reviewer 2 Report

The manuscript “Recent Advances in Characterization of Melanin Pigments in Biological Samples”, by Wakamatsu and Ito, reviews different techniques used to characterize basically eumelanin and pheomelanin in pigmented samples from different animal species.

The manuscript is a bit too technical but it could help readers interested in the field find the most suitable strategy to identify and quantify melanin pigments in different biological samples.

As a rule, figures and tables should be inserted in the manuscript after citing them in the text. Therefore, the authors should revise the manuscript repositioning figures and tables (figure 2, 3, 6, table 1 and 3).

In the conclusions (line 685), the authors discuss measurements “such as colorimeter, solubility, and UV/Visible spectra”. Melanin pigments are known to be insoluble and no technique illustrating the use of solubility measurements is described in the manuscript. Please, correct.

Reviewer 3 Report

-Dear Editor,

I reviewed the mauscript by a Wakamatsu et al., entitled" Recent Advances in Characterization of Melanin Pigments in 2 Biological Samples  " . This manuscript provide interesing information.about the characterization of the melanin. In addition the data are preseted in best quality  and support the context of the manuscript. Moreover, the manuscript are well written and can be published in the present form after a minor check of the pelling of the english langauage.

Many thanks  

Reviewer 4 Report

According to the authors, a recent review on the most commonly applied methods in melanin analysis was made. In the present submitted manuscript, the authors described a set of methodologies for evaluating eumelanin (EM) and pheomelanin (PM) by alkaline hydrogen peroxide oxidation and related methods, as well as by photodegradation methods of EM and PM, and through various physicochemical methods, some of them non-invasive in contrast to the chemical degradative methods.

The manuscript is very important and it is well written, nevertheless I would suggest to compile the information in a Table, where the principles of the methods, advantages and disadvantages should be added, as well as other information that the authors would consider to be important. I do not know, what is the opinion of the authors? I think it would help to make the reading much easier. However, this is only an opinion and the authors can disagree, therefore, this is not mandatory.

Only a very small remark:

The caption of Figure 4 is not clear: was it possible to present PTCA, PTeCA… by the same order for each number? Or is there a valid reason for the presentation made by the authors? In addition, there is a sentence of the text between the Figure and caption.